# LANGUAGE-DRIVEN SEMANTIC SEGMENTATION

**Boyi Li**
Cornell University, Cornell Tech

**Kilian Q. Weinberger**
Cornell University

**Serge Belongie**
University of Copenhagen

**Vladlen Koltun**
Apple

**René Ranftl**
Intel Labs

## ABSTRACT

We present LSeg, a novel model for language-driven semantic image segmentation. LSeg uses a text encoder to compute embeddings of descriptive input labels (e.g., "grass" or "building") together with a transformer-based image encoder that computes dense per-pixel embeddings of the input image. The image encoder is trained with a contrastive objective to align pixel embeddings to the text embedding of the corresponding semantic class. The text embeddings provide a flexible label representation in which semantically similar labels map to similar regions in the embedding space (e.g., "cat" and "furry"). This allows LSeg to generalize to previously unseen categories at test time, without retraining or even requiring a single additional training sample. We demonstrate that our approach achieves highly competitive zero-shot performance compared to existing zero- and few-shot semantic segmentation methods, and even matches the accuracy of traditional segmentation algorithms when a fixed label set is provided. Code and demo are available at https://github.com/isl-org/lang-seg.

## 1 INTRODUCTION

Semantic segmentation is a core problem in computer vision, with the aim of partitioning an image into coherent regions with their respective semantic class labels. Most existing methods for semantic segmentation assume a limited set of semantic class labels that can potentially be assigned to a pixel. The number of class labels is dictated by the training dataset and typically ranges from tens (Everingham et al., 2015) to hundreds (Zhou et al., 2019; Mottaghi et al., 2014) of distinct categories. As the English language defines several hundred thousand nouns (Li et al., 2020c), it is likely that the limited size of the label set severely hinders the potential recognition performance of existing semantic segmentation models.

The main reason for the restricted label sets in existing methods is the cost of annotating images to produce sufficient training data. To create training datasets, human annotators must associate every single pixel in thousands of images with a semantic class label – a task that is extremely labor intensive and costly even with small label sets. The complexity of the annotation rises significantly as the number of labels increases since the human annotator has to be aware of the fine-grained candidate labels. Additionally, inter-annotator consistency becomes an issue when objects are present in an image that could fit multiple different descriptions or are subject to a hierarchy of labels.

Zero- and few-shot semantic segmentation methods have been proposed as a potential remedy for this problem. Few-shot approaches (Shaban et al., 2017; Rakelly et al., 2018; Siam et al., 2019; Wang et al., 2019; Zhang et al., 2019; Nguyen & Todorovic, 2019; Liu et al., 2020b; Wang et al., 2020; Tian et al., 2020; Boudiaf et al., 2021; Min et al., 2021) offer ways to learn to segment novel classes based on only a few labeled images. However, these approaches still require labeled data that includes the novel classes in order to facilitate transfer. Zero-shot methods, on the other hand, commonly leverage word embeddings to discover or generate related features between seen and unseen classes (Bucher et al., 2019; Gu et al., 2020) without the need for additional annotations. Existing works in this space use standard word embeddings (Mikolov et al., 2013) and focus on the image encoder.

In this work, we present a simple approach to leveraging modern language models to increase the flexibility and generality of semantic segmentation models. Our work is inspired by the CLIP model

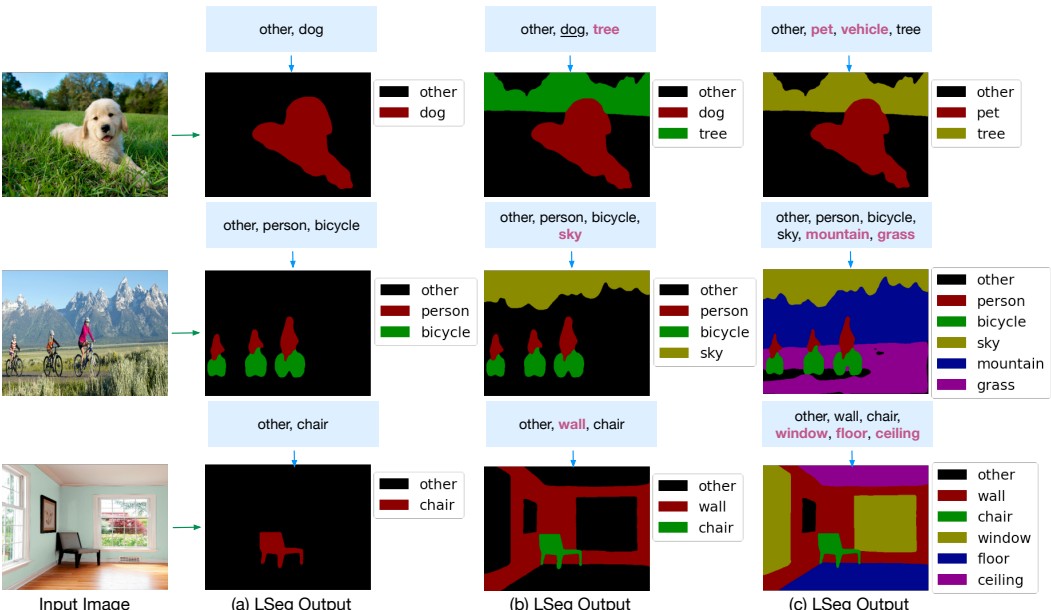

Figure 1: Example results. LSeg is able to handle unseen labels as well as label sets of arbitrary length and order. This enables flexible synthesis of zero-shot semantic segmentation models on the fly. From left to right, labels that are removed between runs are underlined, whereas labels that are added are marked in **bold red**.

for image classification (Radford et al., 2021), which pairs high-capacity image and text encoders to produce robust zero-shot classifiers. We propose to use state-of-the-art text encoders that have been co-trained on visual data, such as CLIP, to embed labels from the training set into an embedding space and to train a visual encoder to produce per-pixel embeddings from an input image that are close to the corresponding label embeddings. Since the text encoder is trained to embed closely related concepts near one another (for example, "dog" is closer to "pet" than to "vehicle"), we can transfer the flexibility of the text encoder to the visual recognition module while only training on the restricted label sets that are provided by existing semantic segmentation datasets. An example is shown in Figure 1 (top row), where the model can successfully label pixels belonging to the class "pet" although the training set did not contain this label.

Our approach enables the synthesis of zero-shot semantic segmentation models on the fly. That is, a user can arbitrarily expand, shrink, or reorder the label set for any image at test time. We further introduce an output module that can spatially regularize the predictions while maintaining this flexibility. We demonstrate several examples of the flexibility of our model in Figure 1. LSeg is able to output different segmentation maps based on the provided label set. For instance, in the last row, output (a) recognizes the chair and identifies all non-chair objects as "other" since these are the only two labels provided to the model. When labels are added, as in (b) and (c), the model is able to successfully segment other objects with the expanded label set.

We conduct quantitative evaluation on a variety of zero- and few-shot semantic segmentation tasks. Our approach outperforms existing methods in zero-shot settings and is competitive across multiple few-shot benchmarks. Unlike the state-of-the-art baselines we compare to, our approach does not require additional training samples. Our experiments also show that introducing the text embeddings incurs only a negligible loss in performance when compared to standard fixed-label segmentation methods.

## 2 RELATED WORK

**Generalized semantic segmentation.** The majority of existing semantic segmentation models are restricted to a fixed label set that is defined by the labels that are present in the training dataset (Minaee et al., 2021). Few-shot semantic segmentation methods aim to relax the restriction of a fixed label set when one or a few annotated examples of novel classes are available at test time. These approaches

learn to find reliable visual correspondences between a query image that is to be labeled and labeled support images that may contain novel semantic classes (Shaban et al., 2017; Rakelly et al., 2018; Siam et al., 2019; Wang et al., 2019; Zhang et al., 2019; Nguyen & Todorovic, 2019; Liu et al., 2020b; Wang et al., 2020; Tian et al., 2020; Wang et al., 2020; Tian et al., 2020; Boudiaf et al., 2021; Min et al., 2021). While this strategy can significantly enhance the generality of the resulting model, it requires the availability of at least one labeled example image with the target label set, something that is not always practical.

Zero-shot semantic segmentation approaches aim to segment unseen objects without any additional samples of novel classes. Text embeddings of class labels play a central role in these works. Bucher et al. (2019) and Gu et al. (2020) propose to leverage word embeddings together with a generative model to generate visual features of unseen categories, while Xian et al. (2019) propose to project visual features into a simple word embedding space and to correlate the resulting embeddings to assign a label to a pixel. Hu et al. (2020) propose to use uncertainty-aware learning to better handle noisy labels of seen classes, while Li et al. (2020b) introduce a structured learning approach to better exploit the relations between seen and unseen categories. While all of these leverage text embeddings, our paper is, to the best of our knowledge, the first to show that it is possible to synthesize zero-shot semantic segmentation models that perform on par with fixed-label and few-shot semantic segmentation methods.

A variety of solutions have been proposed (Zhang et al., 2020b; Liu et al., 2020a; Perera et al., 2020; Zhou et al., 2021) for open-set recognition (Scheirer et al., 2012; Geng et al., 2020). These aim to provide a binary decision about whether or not a given sample falls outside the training distribution, but do not aim to predict the labels of entirely new classes.

Finally, a different line of work explores cross-domain adaptation methods for semantic segmentation by using feature alignment, self-training, and information propagation strategies (Yang et al., 2021; Wang et al., 2021). The target of these works is to enhance the transferability of models to novel visual domains, but they do not address the issue of a restricted label set. As such they are orthogonal to our work.

**Language-driven recognition.** Language-driven recognition is an active area of research. Common tasks in this space include visual question answering (Antol et al., 2015), image captioning (Vinyals et al., 2014), and image-text retrieval (Li et al., 2020a). CLIP (Radford et al., 2021) demonstrated that classic recognition tasks that are not commonly associated with language can strongly benefit from language assistance. CLIP uses contrastive learning together with high-capacity language models and visual feature encoders to synthesize extremely robust models for zero-shot image classification. Recent works have extended this basic paradigm to perform flexible object detection. ViLD (Gu et al., 2021) introduces an advanced zero-shot object detection method that leverages CLIP, whereas MDETR (Kamath et al., 2021) proposes an end-to-end approach that modulates a transformer-based baseline detector with text features that are obtained from a state-of-the-art language model. Like CLIP, these works have shown that the robustness and generality of object detection models can be strongly improved by language assistance. Our work is inspired by these approaches and presents, to the best of our knowledge, the first approach to flexibly synthesize zero-shot semantic segmentation models by leveraging high-capacity language models.

## 3 LANGUAGE-DRIVEN SEMANTIC SEGMENTATION

Our approach, *Language driven Semantic segmentation (LSeg)* embeds text labels and image pixels into a common space, and assigns the closest label to each pixel. We illustrate the framework in Figure 2 and describe each part in detail below.

**Text encoder.** The text encoder embeds the set of $N$ potential labels into a continuous vector space $\mathbb{R}^C$, producing $N$ vectors $T_1, \ldots, T_n \in \mathbb{R}^C$ as outputs (blue vectors in Figure 2). Multiple network architectures are possible, and we use the pretrained Contrastive Language–Image Pre-training (CLIP) throughout (Radford et al., 2021). By design, the set of output vectors is invariant to the ordering of the input labels and allows their number, $N$, to vary freely.

**Image encoder.** Similar to the text encoder, the image encoder produces an embedding vector for every input pixel (after downsampling). We leverage dense prediction transformers (DPT) (Ranftl

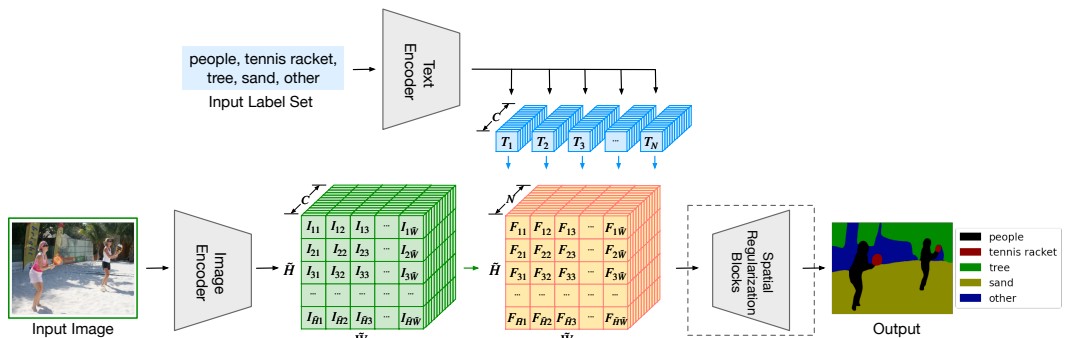

Figure 2: Overview. A text encoder embeds labels into a vector space. An image encoder extracts per-pixel embeddings from the image and correlates the feature of each pixel to all label embeddings. The image encoder is trained to maximize the correlation between the text embedding and the image pixel embedding of the ground-truth class of the pixel. A final spatial regularization block spatially regularizes and cleans up the predictions.

et al., 2021) as the underlying architecture. Assume $H \times W$ is the input image size and $s$ is a user-defined downsampling factor ($s = 2$ in our implementation). We define $\tilde{H} = \frac{H}{s}$, $\tilde{W} = \frac{W}{s}$. The output is a dense embedding $I \in \mathbb{R}^{\tilde{H} \times \tilde{W} \times C}$ (green tensor in Figure 2). We refer to the embedding of pixel $(i, j)$ as $I_{ij}$.

**Word-pixel correlation tensor.** After the image and the labels are embedded, we correlate them by the inner product, creating a tensor of size $\tilde{H} \times \tilde{W} \times N$ (orange tensor in Figure 2), defined as

$$f_{ijk} = I_{ij} \cdot T_k. \tag{1}$$

We refer to the $N$-dimensional vector of inner products between the embedding of pixel $(i, j)$ and all $N$ words as $F_{ij} \in \mathbb{R}^N$, where $F_{ij} = (f_{ij1}, f_{ij2}, ..., f_{ijk})^T$. During training, we encourage the image encoder to provide pixel embeddings that are close to the text embedding of the corresponding ground-truth class. Specifically, given the text embeddings $T_k \in \mathbb{R}^C$ of $N$ labels and the image embedding $I_{ij} \in \mathbb{R}^C$ of pixel $i, j$, we aim to maximize the dot product of the entry $f_{ijk}$ that corresponds to the ground-truth label $k = y_{ij}$ of pixel $i, j$. We achieve this by defining a pixelwise softmax objective over the whole image:

$$\sum_{i,j=1}^{H,W} \text{softmax}_{y_{ij}} \left( \frac{F_{ij}}{t} \right), \tag{2}$$

where $t$ is a user-defined temperature parameter that we set to $t = 0.07$ (Wu et al., 2018; Radford et al., 2021). During training, we minimize a per-pixel softmax with cross-entropy loss (with temperature scaling) as is standard in semantic segmentation[1].

**Spatial regularization.** Due to memory constraints, the image encoder predicts pixel embeddings at lower resolution than the input image resolution. We use an additional post-processing module that spatially regularizes and upsamples the predictions to the original input resolution. During this process, we have to ensure that all operations stay equivariant with respect to the labels. In other words, there should be no interactions between the input channels, whose order is defined by the order of the words and can thus be arbitrary. We evaluate two functions that fulfill this property: a simple cascade of depthwise convolutions (Chollet, 2017) followed by non-linear activations (DepthwiseBlock), and another block that additionally augments the depthwise convolutions with the result of a max-pooling operation over the set of labels (BottleneckBlock) (Li et al., 2019). In a final step we use bilinear interpolation to recover predictions at the original resolution. We refer to these functions as "spatial regularization blocks" and illustrate them in Figure 3.

---

[1] In practice we implement this using the standard nn.CrossEntropyLoss from Pytorch.

**Training details.** We initialize the backbone of the image encoder with the official ImageNet pretrained weights from ViT (Dosovitskiy et al., 2021) or ResNet (He et al., 2016)[2] and initialize the decoder of DPT randomly. During training we freeze the text encoder and only update the weights of the image encoder. We provide the full label set that is defined by each training set to the text encoder for each image.

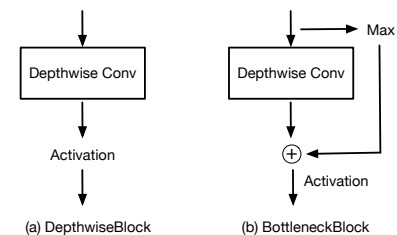

Figure 3: Illustration of BottleneckBlock and DepthwiseBlock.

Our model can be trained on any semantic segmentation dataset and supports flexible mixing of multiple datasets through the text encoder. Existing semantic segmentation models assign a fixed channel in the output to represent the probability of a pixel being the corresponding semantic class. In contrast, our approach can dynamically handle label sets with varying length, content, and order. This property allows synthesizing arbitrary zero-shot semantic segmentation models by simply changing the labels that are fed to the text encoder.

## 4 EXPERIMENTS

We designed LSeg primarily for the zero-shot setting, where labels that are used for inference have never been seen during training. However, due to a lack of a standardized protocol and sufficient datasets and baselines for the zero-shot setting, we compare LSeg to zero- and few-shot semantic segmentation models on few-shot benchmarks. Note that few-shot methods have access to more information and are thus expected to yield higher accuracy. However, the need for labeled samples severely restricts their flexibility compared to our approach.

### 4.1 EXPERIMENTAL SETUP

We follow the protocol of the recent state-of-the-art few-shot method HSNet (Min et al., 2021) and evaluate on three widely-used few-shot semantic segmentation benchmarks: PASCAL-$5^i$ (Everingham et al., 2015), COCO-$20^i$ (Lin et al., 2014), and FSS-1000 (Li et al., 2020c). Following a standard protocol for few-shot segmentation, we use the mean intersection over union (mIoU) and foreground-background intersection of union (FB-IoU) as the evaluation metrics. The mIoU calculates the average IoU over all classes, FB-IoU computes mean value of foreground and background IoUs in fold $i$ and ignores the object classes.

When not stated otherwise we use an LSeg model with the text encoder provided by CLIP-ViT-B/32 and leverage DPT with a ViT-L/16 backbone as the image encoder. For datasets that provide a background or unknown class, we set the corresponding background label to "other". We use SGD with momentum 0.9 and a polynomial learning rate scheduler with decay rate 0.9. We train with a batch size of 6 on six Quadro RTX 6000.

### 4.2 PASCAL-$5^i$ AND COCO-$20^i$

PASCAL-$5^i$ and COCO-$20^i$ are few-shot segmentation datasets that have been created from PASCAL VOC 2012 (Everingham et al., 2015) and the COCO dataset (Lin et al., 2014), respectively. PASCAL-$5^i$ is composed of 20 object classes with corresponding mask annotations and has been evenly divided into 4 folds of 5 classes each. We denote different folds by $5^i$, where $i \in \{0, 1, 2, 3\}$. Similarly, COCO-$20^i$ is composed of 4 folds of 20 classes each.

We compare LSeg to various state-of-the-art few-shot models: OSLSM (Shaban et al., 2017), Co-FCN (Rakelly et al., 2018), AMP-2 (Siam et al., 2019), PANet (Wang et al., 2019), PGNet (Zhang et al., 2019), FWB (Nguyen & Todorovic, 2019), PPNet (Liu et al., 2020b), DAN (Wang et al., 2020), PFENet (Tian et al., 2020), RePRI (Boudiaf et al., 2021), and HSNet (Min et al., 2021). These few-shot methods propose strategies to segment unseen objects based on pretraining on seen categories and finetuning with a few images from the target class. In addition, we also compare

---

[2]We also evaluated on a model initialized with the CLIP image encoder with the same setup and hyperparameters, but observed worse performance than using the ViT initialization.

| Model | Backbone | Method | $5^0$ | $5^1$ | $5^2$ | $5^3$ | mean | FB-IoU |
|---|---|---|---|---|---|---|---|---|
| OSLSM | | 1-shot | 33.6 | 55.2 | 40.9 | 33.5 | 40.8 | 61.3 |
| co-FCN | VGG16 | 1-shot | 36.7 | 50.6 | 44.9 | 32.4 | 41.1 | 60.1 |
| AMP-2 | | 1-shot | 41.9 | 50.2 | 46.7 | 34.7 | 43.4 | 61.9 |
| PANet | ResNet50 | 1-shot | 44.0 | 57.5 | 50.8 | 44.0 | 49.1 | - |
| PGNet | | 1-shot | 56.0 | 66.9 | 50.6 | 50.4 | 56.0 | 69.9 |
| FWB | | 1-shot | 51.3 | 64.5 | 56.7 | 52.2 | 56.2 | - |
| PPNet | | 1-shot | 52.7 | 62.8 | 57.4 | 47.7 | 55.2 | 70.9 |
| DAN | ResNet101 | 1-shot | 54.7 | 68.6 | 57.8 | 51.6 | 58.2 | 71.9 |
| PFENet | | 1-shot | 60.5 | 69.4 | 54.4 | 55.9 | 60.1 | 72.9 |
| RePRI | | 1-shot | 59.6 | 68.6 | **62.2** | 47.2 | 59.4 | - |
| HSNet | | 1-shot | **67.3** | **72.3** | 62.0 | **63.1** | **66.2** | **77.6** |
| SPNet | ResNet101 | zero-shot | 23.8 | 17.0 | 14.1 | 18.3 | 18.3 | 44.3 |
| ZS3Net | | zero-shot | 40.8 | 39.4 | 39.3 | 33.6 | 38.3 | 57.7 |
| LSeg | ResNet101 | zero-shot | **52.8** | **53.8** | **44.4** | **38.5** | **47.4** | **64.1** |
| LSeg | ViT-L/16 | zero-shot | **61.3** | **63.6** | **43.1** | **41.0** | **52.3** | **67.0** |

Table 1: Comparison of mIoU and FB-IoU (higher is better) on PASCAL-$5^i$.

| Model | Backbone | Method | $20^0$ | $20^1$ | $20^2$ | $20^3$ | mean | FB-IoU |
|---|---|---|---|---|---|---|---|---|
| PPNet | | 1-shot | 28.1 | 30.8 | 29.5 | 27.7 | 29.0 | - |
| PMM | ResNet50 | 1-shot | 29.3 | 34.8 | 27.1 | 27.3 | 29.6 | - |
| RPMM | | 1-shot | 29.5 | 36.8 | 28.9 | 27.0 | 30.6 | - |
| RePRI | | 1-shot | 32.0 | 38.7 | 32.7 | 33.1 | 34.1 | - |
| FWB | | 1-shot | 17.0 | 18.0 | 21.0 | 28.9 | 21.2 | - |
| DAN | ResNet101 | 1-shot | - | - | - | - | 24.4 | 62.3 |
| PFENet | | 1-shot | 36.8 | 41.8 | 38.7 | 36.7 | 38.5 | 63.0 |
| HSNet | | 1-shot | **37.2** | **44.1** | **42.4** | **41.3** | **41.2** | **69.1** |
| ZS3Net | ResNet101 | zero-shot | 18.8 | 20.1 | 24.8 | 20.5 | 21.1 | 55.1 |
| LSeg | ResNet101 | zero-shot | **22.1** | **25.1** | **24.9** | **21.5** | **23.4** | **57.9** |
| LSeg | ViT-L/16 | zero-shot | **28.1** | **27.5** | **30.0** | **23.2** | **27.2** | **59.9** |

Table 2: Comparison of mIoU and FB-IoU (higher is better) on COCO-$20^i$.

to the competitive zero-shot baseline ZS3Net (Bucher et al., 2019), which adopts the DeepLabv3+ framework and to Xian et al. (2019) which leverages DeepLabv2. We follow their official code, training setting and training steps on the basis of their provided model pretrained on ImageNet (Deng et al., 2009). We follow the common experimental setup (Nguyen & Todorovic, 2019) and conduct cross-validation over all folds. Assuming that $n_i$ is the number of classes in fold $i$, for each fold $i$, we use images of other folds for training and randomly sampled 1000 images of target fold $i$ for evaluation. We show PASCAL-$5^i$ and COCO-$20^i$ results in Tables 1 and 2. Our model (with the same ResNet101 backbone) outperforms the zero-shot baseline by a considerable margin across folds and datasets and is even competitive with several few-shot methods. We also observe an obvious edge of LSeg by using a larger backbone (ViT-L/16).

## 4.3 FSS-1000

FSS-1000 (Li et al., 2020c) is a recent benchmark dataset for few-shot segmentation. It consists of 1000 object classes with pixelwise annotated segmentation masks. It contains a significant number of unseen or unannotated objects in comparison to previous datasets such as PASCAL and COCO. Following the standard protocol, we split the 1000 classes into training, validation, and test classes, with 520, 240, and 240 classes, respectively. We use a base learning rate of 0.05 and train the model for 60 epochs.

Table 3 compares our approach to state-of-the-art few-shot models. Notably, under the same ResNet101, LSeg could achieve comparative results of the state-of-the-art one-shot method. Also, LSeg even outperforms a state-of-the-art one-shot method: 87.8 mIoU (ours) vs. 86.5 mIoU (HSNet) with a larger backbone ViT-L/16, indicating that LSeg generalizes very well to unseen categories. Figure 4 shows examples of segmentation results on unseen categories.

| Model | Backbone | Method | mIoU |
|---|---|---|---|
| OSLSM | | 1-shot | 70.3 |
| GNet | VGG16 | 1-shot | 71.9 |
| FSS | | 1-shot | 73.5 |
| DoG-LSTM | | 1-shot | 80.8 |
| DAN | ResNet101 | 1-shot | 85.2 |
| HSNet | | 1-shot | 86.5 |
| LSeg | ResNet101 | zero-shot | 84.7 |
| LSeg | ViT-L/16 | zero-shot | **87.8** |

Table 3: Comparison of mIoU on FSS-1000.

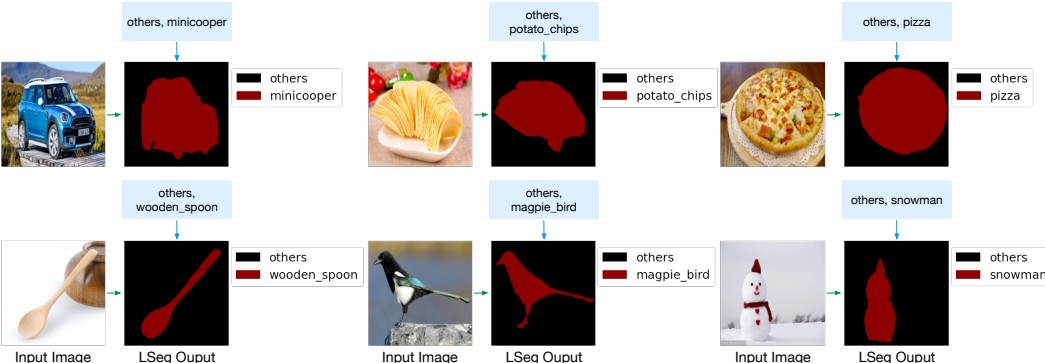

Figure 4: LSeg zero-shot semantic segmentation results on unseen categories of FSS-1000 dataset.

# 5 EXPLORATION AND DISCUSSION

## 5.1 ABLATION STUDIES

We further empirically explore various properties of LSeg. We conduct experiments on the ADE20K dataset (Zhou et al., 2019), which is a standard semantic segmentation dataset that includes a diversity of images and provides pixelwise segmentation of 150 different categories. We set the base learning rate to 0.004 and train the model for 240 iterations. We use SGD with momentum 0.9 and a polynomial learning rate scheduler with decay rate 0.9. We use LSeg with DPT and a smaller ViT-B/32 backbone together with the CLIP ViT-B/32 text encoder unless stated otherwise.

**Spatial regularization blocks.** We first conduct an ablation study on the two variants of the spatial regularization blocks for cleaning up the output. We ablate the different types of blocks as well as stacking various numbers of blocks ($N \in [0, 1, 2, 4]$). The results are shown in Table 4. We notice that a consistent improvement can be achieved by adding a few regularization blocks. The strongest improvement is achieved by stacking two BottleneckBlocks, an addition to the architecture that incurs little overhead.

**Text encoders.** LSeg supports arbitrary text encoders in principle. We show the influence of using different text encoders in Table 5, where we ablate various encoders that are provided by the CLIP

| Block Type | Metric | # depth | | | |
|---|---|---|---|---|---|
| | | 0 | 1 | 2 | 4 |
| DepthwiseBlock | pixAcc [%] | 79.70 | 79.72 | **79.78** | 7.67 |
| | mIoU [%] | 37.83 | 39.19 | **39.45** | 0.18 |
| BottleneckBlock | pixAcc [%] | 79.70 | 79.64 | **79.70** | 79.68 |
| | mIoU [%] | 37.83 | 39.16 | **39.79** | 38.78 |

Table 4: Ablation study on the depth of BottleneckBlock and DepthwiseBlock before the last layer. For both Pixel Accuracy (pixAcc) and mIoU, higher is better. For depth=1, we directly feed the output to reshape without activation.

zero-shot image classification model (Radford et al., 2021). Note that all text encoders feature the same transformer-based architecture that purely operates on text. The main difference between the encoders is the image encoder that was paired during CLIP pretraining (for example, the text encoder denoted by "ViT-B/32" was trained in conjunction with a ViT-B/32 image encoder) and the size of the embedding dimension.

We observe that using RN50×16 achieves the best performance among all text encoders and surpasses the weakest ViT-B/32 text encoder by 2.5%. We conjecture that this is because of the larger size of the embedding that is provided by this encoder.

**Comparison on a fixed label set.** Language assistance helps boost the recognition performance on unannotated or unseen classes. However, there might be a concern that this flexibility hurts the performance on tasks that have a fixed label set. To test this, we train LSeg on ADE20K using the standard protocol on this dataset, where the training and test labels are fixed (that is, there are no unseen class labels at test time). We compare the results to highly competitive standard semantic segmentation models, including OCNet (Yuan et al., 2020), ACNet (Fu et al., 2019), DeeplabV3 (Chen et al., 2017; Zhang et al., 2020a), and DPT (Ranftl et al., 2021). The results are listed in Table 6. We find that LSeg performs competitively when using the RN50 × 16 text encoder and incurs only a negligible loss in performance when compared to the closest fixed-label segmentation method (DPT).

## 5.2 QUALITATIVE FINDINGS

We finally train LSeg on a mix of 7 different datasets (Lambert et al., 2020), including ADE20K (Zhou et al., 2019), BDD (Yu et al., 2020), Cityscapes (Cordts et al., 2016), COCO-Panoptic (Lin et al., 2014; Caesar et al., 2018), IDD (Varma et al., 2019), Mapillary Vistas (Neuhold et al., 2017), and SUN RGBD (Song et al., 2015). Note that we train our model on the original label sets that are provided by these datasets without any preprocessing or relabeling. We follow the same training protocol as on ADE20K and train LSeg with a ViT-L/16 backbone and a ViT-B/32 text encoder for 200 epochs with a base learning rate of 0.004. If there are multiple labels for one class, we only use the first label that is provided during training. We select images from the web and show the results in Figure 5 to illustrate the use of the resulting model.

**Related but previously unseen labels.** We illustrate some salient examples of the capabilities of LSeg to generalize to new classes in Figure 5(a). In the first row, on the left we first start with the label set "sky", "road", "house", and "plant", and observe that the model is capable of segmenting the image into the provided classes. We then change the label "house" to "building" and the label "plant" to "greenery". The model produces a similar segmentation as before on this different but semantically related label set. This is despite the fact that the label "greenery" or even "green" was not present in any of the training images. A similar effect is shown in the second row, where LSeg successfully

| Method | Backbone | Text Encoder (fixed) | embedding dimension | pixAcc [%] | mIoU [%] |
|--------|----------|----------------------|---------------------|------------|----------|
| LSeg | ViT-B/32 | ViT-B/32 | 512 | 79.70 | 37.83 |
| LSeg | ViT-B/32 | ViT-B/16 | 512 | 79.77 | 38.69 |
| LSeg | ViT-B/32 | RN50 × 4 | 640 | 79.85 | 38.93 |
| LSeg | ViT-B/32 | RN50 × 16 | 768 | **80.26** | **40.36** |

Table 5: Ablation study on LSeg with fixed text encoders of different CLIP pretrained models.

| Method | Backbone | Text Encoder | pixAcc [%] | mIoU [%] |
|--------|----------|--------------|------------|----------|
| OCNet | ResNet101 | - | - | 45.45 |
| ACNet | ResNet101 | - | 81.96 | 45.90 |
| DeeplabV3 | ResNeSt101 | - | 82.07 | 46.91 |
| DPT | ViT-L/16 | - | **82.70** | **47.63** |
| LSeg | ViT-L/16 | ViT-B/32 | 82.46 | 46.28 |
| LSeg | ViT-L/16 | RN50 × 16 | **82.78** | **47.25** |

Table 6: Comparison of semantic segmentation results on the ADE20K validation set. For LSeg, we conduct experiments with fixed text encoders of ViT-B/32 and RN50 × 16 CLIP pretrained models.

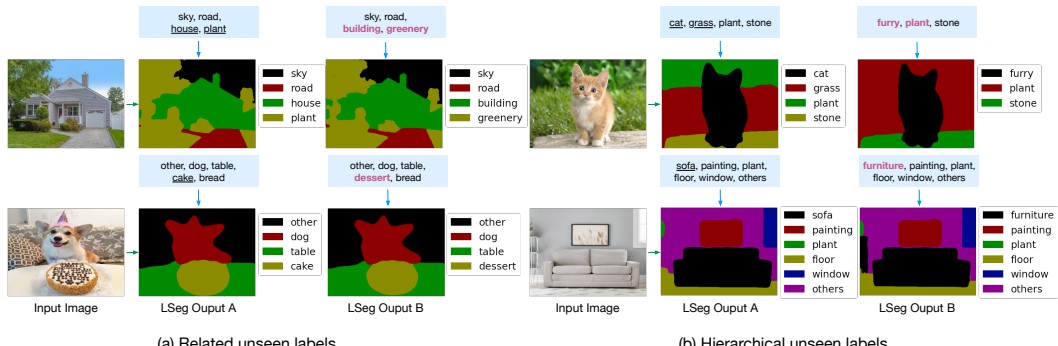

(a) Related unseen labels.  (b) Hierarchical unseen labels.

Figure 5: LSeg examples with related but previously unseen labels, and hierarchical labels. Going from left to right, labels that are removed between runs are underlined, whereas labels that are added are marked in **bold red**.

segments the image and correctly assigns the labels "cake" or "dessert" (again, the label "dessert" was not seen during training), while successfully suppressing the label "bread" which is both visually and semantically related.

**Hierarchical unseen labels.** Figure 5(b) demonstrates that LSeg can implicitly provide correct segmentation maps for hierarchies of labels. In the first row, the model is able to recognize the "cat", "plant" and "grass" segments of the image, as expected since these labels are present in the training set. When replacing "cat" with the label "furry", we notice that the model is able to successfully recognize this parent category (that is, most cats are furry, but not all furry objects are cats). Similarly, when removing the label "grass", we notice that the original "grass" region is merged into "plant", again an indication of an implicit hierarchy that is afforded by the flexibility of the text embeddings. The second row illustrates a similar scenario, where LSeg recognizes the sofa and other objects. However, the small shelf on the left is segmented as the unknown category "other". When we change "sofa" to "furniture", LSeg successfully identifies both the sofa and the small shelf as "furniture". Note that "furniture" never appeared in the training label set.

**Failure cases.** While LSeg in general achieves very promising results, we also observe some failure cases, as illustrated in Figure 6. The left image illustrates that LSeg is only trained with positive samples from a class. When the test-time input labels do not contain any of the true labels for the corresponding pixel, the model assigns the highest probability to the closest label in the text embedding space. In this specific example, the model assigns the label "toy" since the visual features of the dog are apparently closer to "toy" than to "grass" in the embedding space

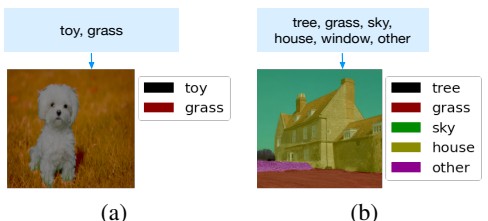

Figure 6: Failure cases.

and there is no other label that can explain the visual features. A second failure case is shown on the right, where the model focuses on a single most likely object when multiple explanations are consistent with the label set. In this specific example, the windows of the house are labeled as "house" instead of window, even thought the label "window" is available as a choice. We hope that these failure cases can inform future work, which could involve augmenting training with negative samples or building fine-grained language-driven semantic segmentation models that can potentially assign multiple labels when multiple explanations fit the data well.

## 6  CONCLUSION

We introduced LSeg, a novel method and architecture for training language-driven semantic segmentation models. LSeg enables a flexible label representation that maps semantically similar labels to similar regions in an embedding space and learns to correlate visual concepts in this space to produce semantic segmentations. Our formulation enables the synthesis of zero-shot semantic segmentation models with arbitrary label sets on the fly. Our empirical results show that the resulting models are strong baselines for zero-shot semantic segmentation and can even rival few-shot segmentation models while not sacrificing accuracy on existing fixed label sets.

## ACKNOWLEDGEMENT

This work was supported in part by the Pioneer Centre for AI, DNRF grant number P1. KQW is supported by grants from the National Science Foundation NSF (IIS-2107161, III-1526012, IIS-1149882, and IIS-1724282), the Cornell Center for Materials Research with funding from the NSF MRSEC program (DMR-1719875), and SAP America.

## ETHICS STATEMENT

We proposed a novel approach to solve the generalized semantic segmentation problem. We use public computer vision datasets and leverage pretrained language models for our experiments. We do not believe that our code or method are inherently subject to concerns of discrimination / bias / fairness, inappropriate potential applications, impact, privacy and security issues, legal compliance, research integrity or research practice issues. However, image datasets and language models may be subject to bias that may be inherited by models trained with our approach.

## REPRODUCIBILITY

Our code is reproducible and can be implemented based on the method description in Section 3 as well as training details in Section 4 and 5. We provide an interactive demo for people to try with input images of their choosing.

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
