# OpenReview forum: "Language-driven Semantic Segmentation"
_ICLR.cc/2022/Conference — ICLR 2022 Poster_

### Official Review · Reviewer_u2M8 · 2021-10-31

**Correctness:** 2
**Technical Novelty And Significance:** 3
**Empirical Novelty And Significance:** 3
**Recommendation:** 5
**Confidence:** 4

**Main Review:**

This method is similar to image classification of CLIP on ImageNet-1k, where the template weights of each class derived from language text, and also called zero-shot learning. As a comparison, this paper v.s. CLIP can be treated as FCN v.s. VGG. I have some questions about this paper
1. In Figure 2, F_{H1} the orange cube is a typo? Should be F_{1W}?
2. In Figure 2, the size is also not correct. The input image H*W*C, the feature should not be that size as stated in the paragraph of bottom of Page 3.
3. In Figure 2, what does C stands for? Input image have C channel, I think C=3 for RGB image. Then Why Text embedding also C channel?
4. In Figure 2, what is the real method to compute training loss? In Equation (2) it computes a global average of softmax_{y_{ij}}(F_{ij})? What does this expression mean? Besides I think an explicit notation of F_{ij} = {f_{ij1}, …f_{ijK}} is necessary.
5. In Table 1, Table 2, Table 3, LSeg with ResNet101 is missing. In Table 6 it said Text Encoder is RN50x16. I think this kind of statement is very strange and an explanation in caption is absolutely necessary.


**Summary Of The Paper:**

This paper is about semantic image segmentation, the template weights of each category is generated by language text.

**Summary Of The Review:**

The clarity of presentation needs improvement. More ablation study necessary.


==========================================

I have read the revised version and still think it is not the final version. For example

1 In Equation(2), what does y_ij means? An image has several labels therefore y_ij is a vector? In this formula it seems it is a scalar.

2 In the revision paper, I still cannot find the training loss. softmax is not a loss objective. A objective function should be something like

                                                                                         min f(x)

3 In comparison, although they are both ResNet-101, but ResNet-101 from CLIP is abosolutely better than other ResNet-101.

===================================

I have read the response and now think the comparison is OK.
From Figure 2 I think every pixel has more than one label. But in Equation (2) it seems y_ij is a scalar.

---

> ### Author Response · Authors · 2021-11-20
> **Response to Reviewer u2M8**
>
> Many thanks for your feedback and suggestions. We have revised the paper accordingly. We address your concerns below.
>
> **Q. “This method is similar to image classification of CLIP on ImageNet-1k.”**
>
> We borrow the simple, but very successful design of CLIP and bring it to semantic segmentation, which is to the best of our knowledge a novel insight. While we believe that our method is distinct from the image classification of CLIP on ImageNet-1k, and want to highlight the differences. CLIP mainly focuses on image classification, while LSeg targets a much harder semantic segmentation problem that interacts each pixel with the text embeddings. LSeg has shown its new capabilities that are not possible with a naive CLIP model.
>
> **Q: “Typos in Figure 2.”**
>
> Thank you for pointing this out. We have corrected the issues in Figure 2.
>
> **Q: “What does C stand for?”**
>
> C is the size of the per-pixel embedding that is extracted by both the image and text feature extractors. To correlate text and image embeddings, they need to be of the same size.
>
> **Q: “What is the real method to compute training loss?”**
>
> We compute the softmax loss for every pixel individually and then average the result over the whole image. Note that this is analogous to how the softmax loss is applied in standard semantic segmentation. We have added the suggested explicit notation.
>
> **Q: “ResNet101 is missing and more ablation studies are necessary.”**
>
> Thank you for this great suggestion. We’ve conducted additional experiments with ResNet101 and find that LSeg still enjoys a strong advantage over previous zero-shot methods. Please see our response to all reviewers for the results of this experiment.
>
> **Q: “Captions in tables missing.”**
>
> We directly use the model name provided from the official CLIP Github and use the corresponding names of text encoders from this code. We’ve added a caption that clarifies this to Table 6.
>
> **Q. “The clarity of presentation needs improvement.”**
>
> Thanks to this point, we have updated the manuscript to improve the clarity. Specifically, we  corrected the issues in Figure 2, added captions in Table 6, added more descriptions in Section 3 and added fair comparisons with ResNet101, etc.
>
> We hope we have addressed your comments satisfactorily and kindly request you to consider increasing your score. Please let us know if there are any new concerns or additional questions we can respond to.

---

> ### Author Response · Authors · 2021-11-26
> **Response to the updated questions of Reviewer u2M8**
>
>
> Thanks for your additional feedback and suggestions. We address your concerns below.
>
> **Q: “In Equation(2), what does y_ij means? An image has several labels therefore y_ij is a vector? In this formula it seems it is a scalar.”**
>
> As stated in the sentence that defines this variable (in Section 3 - Word-pixel correlation tensor), y_ij represents the ground-truth label of pixel i,j, where i and j represent the coordinate of the pixel.
>
> **Q. “Cannot find the training loss”**
>
> Thanks for your feedback for improving clarity. We only now realize that the naming “softmax” might be confusing here. What we mean is that we minimize a per-pixel softmax with cross-entropy loss (with temperature scaling) as is standard in semantic segmentation. We are not allowed to update the paper at this point due to ICLR policy, but we will add the full definition of the loss to the camera-ready version. In practice we implement this using the standard `nn.CrossEntropyLoss` from Pytorch (https://pytorch.org/docs/stable/generated/torch.nn.CrossEntropyLoss.html).
>
> **Q. “In comparison, although they are both ResNet-101, but ResNet-101 from CLIP is abosolutely better than other ResNet-101.”**
>
> This might be a misunderstanding. As has been mentioned in the paper (Section 3 - Training details), we select DPT with a ResNet-101 backbone as the image encoder, not the CLIP pretrained image encoder. We initialize the backbone of the image encoder with the official ImageNet pretrained weights. This means, we use the same ResNet-101 backbone as the other methods, therefore leading to a fair comparison. We will clarify this point in the camera-ready version as well.
>
> In light of our clarifications and completion of many requested experiments, we would like to ask if you are willing to reconsider your score, and also if there are any new concerns or additional questions we can respond to!

---

> ### Author Response · Authors · 2021-11-29
> **Thanks for the review and has the updated response addressed your concerns?**
>
> Dear reviewer,
>
> Thanks for your additional feedback and helpful suggestions. We have posted a response to your updated comments. We would like to ask if it has addressed your concerns?
>
> In light of our clarifications and completion of many requested experiments, we would like to ask if you are willing to reconsider your score, and also if there are any new concerns or additional questions we can respond to!

---

> ### Author Response · Authors · 2021-11-30
> **Glad to hear the response addressed your concerns and we address your questions here**
>
> Thanks for your updated review and detailed explanation! We apologize that we might make the equation hard to understand, we address your questions here.
>
> **Q: “From Figure 2 I think every pixel has more than one label. But in Equation (2) it seems y_ij is a scalar.”**
>
> From Figure 2, after feeding an image into the image encoder, we could obtain $C \times \widetilde H \times  \widetilde W$ image features (the green cube). For each pixel, it has $C$ dimensions, we reshape this cube into $C \times (\widetilde H \times \widetilde W)$. To interact the text features with image features, we element-wise multiply the image features $C \times (\widetilde H \times \widetilde W)$ with text features $C \times N$. Then we would get a $N \times (\widetilde H \times \widetilde W)$ vector, we reshape this vector into $N \times \widetilde H \times \widetilde W$ (the orange cube). And for each pixel, it has $N$ dimensions, each element represents the corresponding probability for each label (${1,2,..,N}$). For the loss function, we minimize a per-pixel softmax with cross-entropy loss (with temperature scaling) as is standard in semantic segmentation. In this way, as displayed in Equation (2), each $y_{ij}$ is a scalar in our loss function that represents the groundtruth label (${1,2,..,N}$). We will reorganize the statement in our paper in detail.
>
> Thanks again for your helpful feedback. We are happy our previous response could address your concerns and we will definitely incorporate these useful comments into our camera ready version! We hope we have addressed your comments satisfactorily and kindly request you to consider increasing your score. Please let us know if there are any new concerns or additional questions we can respond to!

---

> > ### Author Response · Authors · 2021-12-01
> > **We are happy to answer any of your additional comments**
> >
> > Dear Reviewer u2M8,
> >
> > Thanks for your time and patience. When you get a chance to look at our updated response, we hope we have addressed your comments satisfactorily. And we would like to let you know that we are willing and happy to answer your additional comments if you have any.

---

### Official Review · Reviewer_3LsU · 2021-11-01

**Correctness:** 4
**Technical Novelty And Significance:** 3
**Empirical Novelty And Significance:** 3
**Recommendation:** 8
**Confidence:** 3

**Main Review:**

The biggest strength of the proposed approach is its simplicity and its flexibility since it can dynamically handle arbitrary label sets on the fly with varying length, content, and order.

Weaknesses:
1.	While the performance on FSD-1000 and ADE-20k are impressive, it is compared against just 1-shot. Also, the gap is larger in Pascal-5i and COCO-20i with less mIoU and FB-IoU for LSeg.
2.	The strongest baseline uses ResNet-101 which has much fewer parameters (~45M) than the backbone used by LSeg (307M). So, it does not seem like a fair comparison.



**Summary Of The Paper:**

The paper proposes Language driven Semantic segmentation (LSeg) for semantic segmentation. Essentially, LSeg embeds text labels and image pixels into a common space, and assigns the closest label to each pixel. LSeg is flexible and can dynamically handle arbitrary label sets on the fly with varying length, content, and order. The paper demonstrates that LSeg achieves comparable performance as state of the art few-shot semantic segmentation networks on FSD-1000 even though it is used in zero-shot setting. When a fixed label set is used based on the data set (all labels in the training set), it also matches the accuracy of traditional segmentation algorithms on ADE-20k. LSeg uses text encoder from CLIP-ViT-B/32 which is frozen during training while the weights of the image decoder (DPT with a ViT-L/16) is updated to maximize the correlation between the text embedding and the image pixel embedding of the ground truth class of the pixel. Spatial regularization is applied at the end that also up samples the predictions to the original input resolution.

**Summary Of The Review:**

Experiments section is weak. Most of the tables contain much weaker and older models for comparison (even VGG16). The biggest concern is that ViT-L has 307M parameters while the largest backbone used in comparisons is ResNet-101 which has about 45M parameters. ViT-B has 86M which is still larger than ResNet-101 but will at least be a fairer comparison that using ViT-L for image decoder.

However, the flexibility of the approach is powerful. Lseg can dynamically handle arbitrary label sets on the fly with varying length, content, and order.

Hence the recommended rating is “marginally above” the acceptance threshold.

=====

The reviewer thanks the authors for their response. Glad to see that LSeg still shows improvements when using ResNet101 as backbone. Updated the rating.

---

> ### Author Response · Authors · 2021-11-20
> **Response to Reviewer 3LsU**
>
>
> Thank you for your helpful and positive feedback on the flexibility and power of our approach. We address your concerns below.
>
> **Q: “While the performance on FSD-1000 and ADE-20k are impressive, it is compared against just 1-shot. Also, the gap is larger in Pascal-5i and COCO-20i with less mIoU and FB-IoU for LSeg.”**
>
> FSS-1000 is a relatively new dataset which contains 1000 classes and a significant number of objects that have never been seen or annotated in previous datasets. We chose this as an additional experiment to showcase that LSeg is able to handle a very large label set well. Please note that the one-shot setting is considerably easier than the zero-shot setting, as even a single sample provides more context for unseen classes than having no anchor at all. As such, these baselines have an inherent advantage over LSeg. It is somewhat surprising that LSeg is still able to provide competitive performance here. Also please note that for Pascal-5i and COCO-20i, LSeg shows a considerable improvement over other zero-shot methods (i.e. methods that do not have an inherent advantage).
>
> Regarding ADE-20k, it might be a misunderstanding, there is no few-shot experiment on ADE-20k available. We conduct ablation studies on ADE-20k to empirically explore various properties of LSeg.
>
> **Q: “Fair comparison.”**
>
> Thank you for bringing this up. We conducted additional experiments with a ResNet101 backbone and the results are consistent with the original submission. Please have a look at our response to all reviewers for detailed results of these experiments.
>
> We hope we could address your concerns satisfactorily and kindly request you to consider increasing your score. Please let us know if there are any new concerns or additional questions we can respond to.

---

### Official Review · Reviewer_Cogy · 2021-11-02

**Correctness:** 4
**Technical Novelty And Significance:** 2
**Empirical Novelty And Significance:** 3
**Recommendation:** 6
**Confidence:** 3

**Main Review:**

Strength:

- This paper proposes to use language knowledge (especially semantic similarities, I guess) to label the pixels of new classes for zero-shot semantic segmentation. The idea of distilling knowledge from pre-trained models is reasonable and becomes a new trend for many zero-shot settings.
- The experiment results seem good, which outperforms the current SOTA on a zero-shot setting and is on par with some few-shot results.

Weaknesses:

- The method is not that novel. It just computes the semantic similarity between the language embedding from CLIP and visual embedding from DPT, and adopts cross-entropy as supervision. Such a simple design cannot address some critical issues in zero-shot semantic segmentation, as mentioned in Failure cases in Sec. 5.1. The authors notice the issues but leave them to the readers.

- The result comparison in the experiment is not fair. This work uses a stronger backbone (ViT) in the image encoder than the previous works (RN-101). Is there any way to get a fair comparison?

**Summary Of The Paper:**

This paper uses the pre-trained large model (CLIP) to transfer the language knowledge to the unseen labels for zero-shot semantic segmentation. The idea is reasonable and coherent with previous works that distill knowledge from pre-trained models. The experiments also show good results on several benchmarks in a zero-shot setting. However, the method is not novel and leads to many problems.

**Summary Of The Review:**

I like the idea of utilizing pre-trained models to transfer the knowledge for zero-shot semantic segmentation, and I think it would be the main trend for many other zero-shot settings. However, this paper only adopts a naive method to solve the problem, leaving some critical issues to the readers. So my initial rating is borderline.

---

> ### Author Response · Authors · 2021-11-20
> **Response to Reviewer Cogy**
>
> Thank you for your helpful and positive feedback! We address your concerns below.
>
> **Q: “The method is not that novel.”**
>
> We borrow the simple, but very successful design of CLIP and bring it to semantic segmentation, which is to the best of our knowledge a novel insight. Also, CLIP mainly focuses on image classification, however, LSeg aims to solve a much harder semantic segmentation problem that interacts each pixel with the text embeddings. LSeg has shown its new capabilities that are not possible with a naive CLIP model.
>
> **Q: “Simple design cannot address critical issues.”**
>
> Please note that existing methods suffer from more serious similar issues (restricting in the desired label set). However, LSeg presents a clear advance to dynamically handle arbitrary label sets on the fly with varying length, content, and order. To stimulate further research, we also show and discuss limitations and failure cases.
>
> **Q: “Fair comparison.”**
>
> Thank you for bringing this up. We conducted additional experiments with a ResNet101 backbone and the results are consistent with the original submission. Please have a look at our response to all reviewers for detailed results of these experiments.
>
> We hope we could address your concerns satisfactorily and kindly request you to consider increasing your score. Please let us know if there are any new concerns or additional questions we can respond to.

---

> ### Author Response · Authors · 2021-11-29
> **Thanks for the review and has the response addressed your concerns?**
>
> Dear reviewer,
>
> Thank you for your helpful and positive feedback. We are happy that you recognize the flexibility and potential powerfulness of our approach. We have posted a response to your comments. We would like to ask if it has addressed your concerns?
>
> In light of our clarifications and completion of many requested experiments, we would like to ask if you are willing to reconsider your score, and also if there are any new concerns or additional questions we can respond to!

---

> > ### Comment · Reviewer_Cogy · 2021-11-29
> > **Thanks for your reply and demos. Still concern about the values in the research area.**
> >
> > I really appreciate the authors comparing their results with other works with the same backbone network, and further showing their method with a video. I am still worried that this paper seems more like an engineering work of applying CLIP to semantic segmentation. A critical issue of this task might be how to distinguish similar categories with the help of CLIP (e.g., the base categories include 'dog', but the test image contains both 'dog' and similar class 'cat'). My overall rating is positive and I think borderline acceptance is a reasonable score.

---

> > > ### Author Response · Authors · 2021-11-29
> > > **Thank you for your positive feedback and we address your concerns here**
> > >
> > > Thank you for your helpful and positive comments. We address your concerns here.
> > >
> > > **Q: “Differences from CLIP.”**
> > > 1) In the experiments of our paper (Section 3 - Training details), we only use the CLIP text encoder. This means, our model doesn’t rely on the pretrained CLIP image encoder. For image encoder, we select DPT as image encoder. We initialize the backbone of the image encoder with the pretrained weights from ViT or ResNet and initialize the decoder of DPT randomly.
> > >
> > > 2) We provide a method on how to interact labels with images for semantic segmentation, a much harder pixel-level recognition task, which cannot be achieved by a naive CLIP model or previous semantic segmentation approaches. And the text encoder and image encoder could not be restricted in CLIP.
> > >
> > > 3) LSeg deals with semantic segmentation and shows new capabilities that are not possible with a naive CLIP model. For example, in Figure 5 (b) 2nd row of the paper and the demo, when we replace the `sofa` with the `furniture`, LSeg not only could recognize the sofa but also the stand. It is impossible for an image classification model to achieve this new capability.
> > >
> > > **Q: “How to distinguish similar categories with the help of CLIP.”**
> > >
> > > 1) As has been indicated in Table 6, LSeg shows comparative results with the highly competitive fixed-label segmentation methods (the fixed-label set contains similar categories). Also, regarding the similar categories, LSeg is still able to distinguish `cat` and `dog` very well, please check an example here: https://www.youtube.com/watch?v=0XDGuirDj5g.
> > >
> > > 2) We understand that the reviewer has concerns regarding the failure cases. However, this issue could be easily solved by adding the label `toy` into training. So during testing, the model is able to distinguish between `dog`, `other` and `toy`. Also, please note that existing methods require additional transferring functions and could segment the whole image as `toy`.
> > >
> > > **Q: “Values in the research area.”**
> > >
> > > Regarding the value in the research area, we would like to clarify several points below.
> > >
> > > 1) As far as we know, LSeg is the first approach to flexibly synthesize zero-shot semantic segmentation models by leveraging high-capacity language models.
> > >
> > > 2) We totally agree with what has been mentioned in the reviewer’s first review, we also believe that LSeg provides a way or direction that would be the main trend for many other zero-shot settings.
> > >
> > > 3) We compare our methods with state-of-the-art few-shot methods using our zero-shot setting and witness the clear advantage of LSeg on zero-shot semantic segmentation. To be noted, in Table 3, we show the comparison with FSS-1000. FSS-1000 is a relatively new dataset which contains 1000 classes and a significant number of objects that have never been seen or annotated in previous datasets. We chose this as an additional experiment to showcase that LSeg is able to handle a very large label set well, which is very hard for previous zero-shot methods. LSeg has no anchor at all during training, so it is somewhat surprising that LSeg is still able to provide competitive performance here (even using the same ResNet101 backbone).
> > >
> > > 4) Existing methods often require an additional manual transferring function to apply the model trained from one dataset to another. Since LSeg can dynamically handle arbitrary label sets on the fly with varying length, content, and order. Therefore, LSeg doesn’t require any additional manual transferring function in this aspect.
> > >
> > > 5) LSeg is simple yet efficient. It is somewhat surprising that LSeg can dynamically handle arbitrary label sets on the fly with varying length, content, and order very well. We hope LSeg could provide a simple but solid baseline to boost future research on semantic segmentation and related zero-shot problems.
> > >
> > > We hope we could address your concerns satisfactorily. Please let us know if there are any new concerns or additional questions we can respond to!

---

> > > > ### Author Response · Authors · 2021-12-01
> > > > **We are happy to answer any of your additional comments**
> > > >
> > > > Dear Reviewer Cogy,
> > > >
> > > > Thanks for your time and patience. When you get a chance to look at our updated response, we hope we have addressed your comments satisfactorily. And we would like to let you know that we are willing and happy to answer your additional comments or concerns if you have any.

---

### Author Response · Authors · 2021-11-20
**Response to all reviewers and additional experimental results for fair comparison**

We thank the reviewer for their helpful comments. We are happy that the reviewers recognize the flexibility and potential powerfulness of our approach. We revised the paper according to their comments, please check the updated version for details.

Specifically:

1. We added additional experiments with a ResNet101 backbone for the Pascal-5i, COCO-20i, and FSS-1000 evaluations. The experiments show that the difference in performance is not solely due to a larger backbone.
2. We revised Figure 2 to be consistent with the text.
3. We added clarification about the notation.

**Video demo**

We additionally created a short video demo to further showcase the capabilities of LSeg which can be found here: https://www.youtube.com/watch?v=GqtMNeCsRqw .

*Additional experimental results for fair comparison*

We answer a common question of all reviewers here and address other comments in the responses to individual reviewers below.

Q: "Missing comparison with ResNet101 backbone."

Thank you for bringing this up. We have conducted additional experiments with a ResNet101 backbone and added these experiments to the paper. The results can also be found in Table Table 1, Table 2, and Table 3 below. We observe the same trend as before: LSeg, even with the smaller backbone, clearly outperforms state-of-the-art zero-shot methods on Pascal-5i and COCO-20i and is competitive to one-shot methods on FSS-1000.

Table 1: Comparison of mIoU and FB-IoU (higher is better) on PASCAL-5i

| Model  | Backbone  | Method    | $5^0$  | $5^1$  | $5^2$  | $5^3$  | mean   | FB-IoU |
|--------|-----------|-----------|--------|--------|--------|--------|--------|--------|
| SPNet  | ResNet101 | zero-shot | 23.8   | 17.0   | 14.1   | 18.3   | 18.3   | 44.3   |
| ZS3Net | ResNet101 | zero-shot | 40.8   | 39.4   | 39.3   | 33.6   | 38.3   | 57.7   |
| LSeg   | ResNet101 | zero-shot | $52.3$ | $53.8$ | $44.4$ | $38.5$ | $47.3$ | $64.1$ |
| LSeg   | ViT-L/16  | zero-shot | $61.3$ | $63.6$ | $43.1$ | $41.0$ | $52.3$ | $67.0$ |

Table 2: Comparison of mIoU and FB-IoU (higher is better) on COCO-20i

| Model  | Backbone  | Method    | $20^0$  | $20^1$  | $20^2$  | $20^3$  | mean   | FB-IoU |
|--------|-----------|-----------|--------|--------|--------|--------|--------|--------|
| ZS3Net | ResNet101 | zero-shot | 18.8   | 20.1   | 24.8   | 20.5   | 21.1   | 55.1   |
| LSeg   | ResNet101 | zero-shot | $22.1$ | $25.1$ | $24.9$ | $21.5$ | $23.4$ | $57.9$ |
| LSeg   | ViT-L/16  | zero-shot | $28.1$ | $27.5$ | $30.0$ | $23.2$ | $27.2$ | $59.9$ |


Table 3: Comparison of mIoU and FB-IoU (higher is better) on FSS-1000

| Model  | Backbone  | Method    | FB-IoU |
|--------|-----------|-----------|--------|
| HSNet  |   VGG16   |  1-shot   |  70.3  |
| HSNet  |   VGG16   |  1-shot   |  71.9  |
| HSNet  |   VGG16   |  1-shot   |  73.5  |
| HSNet  |   VGG16   |  1-shot   |  80.8  |
| DAN    | ResNet101 |  1-shot   |  85.2  |
| HSNet  | ResNet101 |  1-shot   |  86.5  |
| LSeg   | ResNet101 | zero-shot |  84.7  |
| LSeg   | ViT-L/16  | zero-shot | $87.8$ |

Note: As has been mentioned in the paper (Section 3 - Training details), we select DPT with a ResNet-101 backbone as the image encoder, not the CLIP pretrained image encoder. We initialize the backbone of the image encoder with the official ImageNet pretrained weights. This means, we use the same ResNet-101 backbone as the other methods, therefore leading to a fair comparison. We will clarify this point in the camera-ready version.

We hope we could address your concerns satisfactorily and kindly request you to consider increasing your score. Please let us know if there are any new concerns or additional questions we can respond to.

---

### Public Comment · ~Yaxing_Wang2 · 2022-01-30
**Zero-shot setting**

Thank authors for providing vary interesting paper. I have concerns which are as following:
1) What is the loss function for zero-shot, since  the paper indicates that it is the cross entropy, which I think needs ground truth. 2) extensive ablations are conducted, which seems are about the fully supervised setting. However, this paper highlights zero-shot setting, is the ablation study suitable for zero-shot. 3) Do authors plan to release the zero-shot learning code, since I have trained the provided code, which requires GT.

---

> ### Public Comment · ~Boyi_Li1 · 2022-01-31
> **Response**
>
> Hi Yaxing,
>
> Thanks for your questions. We address your concerns below.
>
> **Q. “What is the loss function for zero-shot, since the paper indicates that it is the cross-entropy, which I think needs ground truth.”**
>
> There might be some misunderstanding in our pipeline. We use cross-entropy loss with labels of certain classes during training. We claim our method as zero-shot segmentation in the sense that our model is able to segment classes that are ‘not seen’ during training.
>
>
> **Q. “Extensive ablations are conducted, which seems to be about the fully supervised setting.”**
>
> Different from previous methods, the biggest strength of the proposed approach is its simplicity and its flexibility since it can dynamically handle arbitrary label sets on the fly with varying length, content, and order. We show the ability of LSeg on zero-shot settings. Language assistance helps boost the recognition performance on unannotated or unseen classes. However, there might be a concern that this flexibility could hurt the performance of tasks that have a fixed label set. To test this, we train LSeg on ADE20K using the standard protocol on this dataset for ablation study, so we would like to compare its ability when training with a fully supervised setting, using different settings. This has also been mentioned in Section 5.2 of the paper.
>
>
> **Q “Do authors plan to release the zero-shot learning code since I have trained the provided code, which requires GT.”**
>
> Yes. And we already released our demo and ablation study in https://github.com/isl-org/lang-seg. Regarding the demo, you could also try various input labels that haven’t been seen during training. For Table 1, 2, 3, we strictly follow the label split from HSNet as mentioned in the paper, you could follow it to check the detailed splits and replace the label set based on your experimental requirements as well. We will also keep updating our code, please check the GitHub for more details.
>
> We hope we have addressed your comments satisfactorily. Please feel free to email us if you have any further questions.

---

### Decision · Program_Chairs · 2022-01-20

**Decision:**

Accept (Poster)

**Comment:**

The paper presents an approach to semantic segmentation based on text embedding of class labels. This enables zero-shot semantic segmentation with class labels that were not seen during training. I appreciate the new ablation against a ResNet-101 backbone. I don't find the similarity with CLIP substantial, and I recommend that the paper is accepted.